# Unveiling the Yin-Yang Balance of M1 and M2 Macrophages in Hepatocellular Carcinoma: Role of Exosomes in Tumor Microenvironment and Immune Modulation

**DOI:** 10.3390/cells12162036

**Published:** 2023-08-10

**Authors:** Stavros P. Papadakos, Nikolaos Machairas, Ioanna E. Stergiou, Konstantinos Arvanitakis, Georgios Germanidis, Adam Enver Frampton, Stamatios Theocharis

**Affiliations:** 1First Department of Pathology, School of Medicine, National and Kapodistrian University of Athens, 10679 Athens, Greece; stavrospapadakos@gmail.com; 2Second Department of Propaedeutic Surgery, National and Kapodistrian University of Athens, Laikon General Hospital, 11527 Athens, Greece; nmachair@med.uoa.gr; 3Pathophysiology Department, School of Medicine, National and Kapodistrian University of Athens, 11527 Athens, Greece; stergiouioanna@hotmail.com; 4Division of Gastroenterology and Hepatology, First Department of Internal Medicine, AHEPA University Hospital, Aristotle University of Thessaloniki, St. Kiriakidi 1, 54636 Thessaloniki, Greece; arvanitak@auth.gr (K.A.); georgiosgermanidis@gmail.com (G.G.); 5Basic and Translational Research Unit (BTRU), Special Unit for Biomedical Research and Education (BRESU), Faculty of Health Sciences, School of Medicine, Aristotle University of Thessaloniki, 54636 Thessaloniki, Greece; 6Department of Surgery & Cancer, Imperial College London, Hammersmith Hospital, London W12 0NN, UK; 7Oncology Section, Surrey Cancer Research Institute, Department of Clinical and Experimental Medicine, FHMS, University of Surrey, The Leggett Building, Daphne Jackson Road, Guildford GU2 7WG, UK; 8HPB Surgical Unit, Royal Surrey NHS Foundation Trust, Guildford GU2 7XX, UK

**Keywords:** exosomes, extracellular vesicles, hepatocellular carcinoma (HCC), immunotherapy, macrophages, tumor microenvironment (TME)

## Abstract

Hepatocellular carcinoma (HCC) is a primary liver cancer with a high mortality rate and limited treatment options. Recent research has brought attention to the significant importance of intercellular communication in the progression of HCC, wherein exosomes have been identified as critical agents facilitating cell-to-cell signaling. In this article, we investigate the impact of macrophages as both sources and targets of exosomes in HCC, shedding light on the intricate interplay between exosome-mediated communication and macrophage involvement in HCC pathogenesis. It investigates how exosomes derived from HCC cells and other cell types within the tumor microenvironment (TME) can influence macrophage behavior, polarization, and recruitment. Furthermore, the section explores the reciprocal interactions between macrophage-derived exosomes and HCC cells, stromal cells, and other immune cells, elucidating their role in tumor growth, angiogenesis, metastasis, and immune evasion. The findings presented here contribute to a better understanding of the role of macrophage-derived exosomes in HCC progression and offer new avenues for targeted interventions and improved patient outcomes.

## 1. Introduction

Hepatocellular carcinoma (HCC) poses a significant global health burden [1]. HCC constitutes nearly 90% of all liver cancer cases, while cholangiocarcinoma (CCA) accounts for approximately 10% [2,3]. According to Global Cancer Statistics (GLOBOCAN), liver cancer ranks as the sixth most prevalent cancer type and ranks fourth in terms of cancer-related fatalities [4]. In 2020, it was estimated that there would be around 906,000 newly diagnosed cases and approximately 830,000 fatalities due to this disease [4]. The highest risk of developing liver cancer was observed in middle Socio-demographic Index (SDI) countries, with approximately 1 in 26 men and 1 in 76 women being affected, while the lowest risk was observed in low-middle SDI countries, with approximately 1 in 93 men and 1 in 195 women being affected [5]. The increase in liver cancer cases from 732,000 (95% UI: 702.000–747.000) in 2006 to 1.0 million (95% UI: 953.000–1.0 million) in 2016 was primarily driven by population aging and population growth [5]. Among the total 38% rise in cases from 2006 to 2016, approximately 16% can be attributed to the aging of the population, 12% to population growth, and 9% to an elevation in age-specific incidence rates [5]. Major risk factors for the development of cirrhosis and hence HCC include chronic infection with the hepatitis B (HBV) or C (HCV) virus, excessive alcohol intake, and non-alcoholic fatty liver disease (NAFLD) [6]. Other risk factors include exposure to aflatoxin, diabetes, obesity, and smoking [7]. Due to advanced disease at diagnosis or poor patient performance status, only a small proportion of patients with HCC are candidates for curative treatment options [8,9,10]. Despite recent scientific achievements, the 5-year survival rate for HCC remains dismal at approximately 20% [11].

The role of tumor-associated macrophages (TAMs) is influenced by multiple layers of signals from the tumor microenvironment (TME) in the co-evolving cancer ecosystem [12,13,14]. These signals include hypoxia, inflammation, and immune suppression. Hypoxia is a condition where there is a lack of oxygen in the TME, which can activate hypoxia-inducible factors (HIFs) that promote TAM recruitment and polarization towards a pro-tumor phenotype [15]. Inflammation serves as an additional signal that can trigger the activation of TAMs through the release of cytokines and chemokines by tumor cells or other immune cells present in the TME [16]. Immune suppression is also a key factor that shapes TAM phenotypic states, as TAMs can inhibit anti-tumor immune responses through various mechanisms such as antigen presentation and T cell suppression [16,17]. Exosomes are small extracellular vesicles that play a role in immune response, cell communication, and intercellular regulation. They carry proteins, nucleic acids, and lipids that influence the activity of recipient cells [18]. Exosomes play a crucial role in facilitating intercellular communication and signaling within the TME of HCC [19,20]. The TME refers to the complex network of cells and molecules that surround and support tumor growth [21]. Exosomes can modulate immune responses, mediate tumor-stroma crosstalk, serve as biomarkers for diagnosis and prognosis, and offer new therapeutic targets for HCC [22,23].

## 2. Understanding the Biology and Biogenesis of Exosomes

Extracellular vesicles (EVs) are composed of two main subsets: exosomes and ectosomes, or microvesicles [24,25,26,27]. Exosomes typically range in size from 40 to 150 nm, while microvesicles can vary in size from 50 to 1000 nm [24,28]. EVs are enriched with tetraspanins such as CD9, CD63, CD81, syntenin, integrins, the programmed cell death 6 interacting protein (also known as Alix), TSG101, and flotillin [29,30]. However, there is heterogeneity in the expression of some EV biomarkers among different cell types, and there is overlap in size and protein expression between ectosomes and exosomes. Exosomes are enriched with CD63, CD9, CD81, Alix, and syntenin [19,25]. Microvesicles are formed through budding at the plasma membrane and are believed to be enriched with CD9 and CD81 [24]. It is challenging to isolate pure populations of microvesicles or exosomes due to their overlapping protein marker expression and size [31]. Exosomes have been extensively studied in various types of malignancies [19,32,33,34,35].

### 2.1. Biogenesis of Exosomes

Exosome biogenesis: The process of exosome biogenesis involves two main steps: the inward budding of membranous vesicles within endosomes and their subsequent release into multivesicular bodies (MVBs) [36,37]. As early endosomes mature into late endosomes, MVBs form and accumulate intraluminal vesicles. Following maturation, MVBs are directed for fusion with either lysosomes, leading to cargo degradation, or the plasma membrane, resulting in the release of their contents into the extracellular space. During the process of exosome biogenesis, transmembrane proteins are incorporated into the invaginating membrane, maintaining a comparable topological arrangement to that of the plasma membrane [28].

The composition of cargo: The content of exosomes differs based on the specific cell type [37]. As per the most recent update from the exosome content database (Exocarta), exosomes originating from various organisms and cell types have been identified to contain 4563 proteins, 194 lipids, 1639 mRNAs, and 764 miRNAs [38]. The protein content of exosomes is largely influenced by their cellular origin and is enriched in specific molecules such as tetraspanins, lactadherin, integrins, cytoplasmic enzymes, chaperones, membrane trafficking proteins, proteins involved in MVB formation, cytoskeletal proteins, and signal transduction proteins [25]. The conformation of exosome-specific proteins may vary based on the cell type or tissue of origin and can also be influenced by physiological changes and cellular stimulation [25]. For instance, exosomes derived from antigen-presenting cells are enriched in antigen-presenting molecules [39], while tumor-derived exosomes contain tumor antigens and immunosuppressive proteins [40]. Additionally, exosomes carry lipids such as cholesterol, diglycerides, glycerophospholipids, phospholipids, and sphingolipids or glycosylceramides, which play crucial roles in lipid metabolism and act as carriers of bioactive lipids [41]. The fatty acids (i.e., docosahexaenoic acid and lysophosphatidylcholine) found in exosomes can enhance the antigenic capacity of dendritic cells, whereas exosomes rich in prostaglandin PGE2 are implicated in tumor immune evasion and tumor growth promotion [42]. In addition to proteins and lipids, exosomes contain functional RNA molecules, including mRNAs, microRNAs (miRNAs), and long non-coding RNAs (lncRNAs) [43,44,45]. These exosomal RNAs, particularly miRNAs, have demonstrated functional roles in recipient cells [33,34].

Although the influence of other exosomal contents on recipient cell behavior should not be underestimated, miRNAs assume crucial roles in numerous processes. Certain exosomal pathways have been observed to remove tumor-suppressor miRNAs, which impede metastatic progression [46]. Meanwhile, exosomal miRNAs have been extensively implicated in tumor promotion [47,48].

Cargo Sorting: The precise mechanisms involved in the sorting of proteins and lipids into exosomes are not fully understood [25]. However, several potential mechanisms have been proposed, including the involvement of heteromeric protein complexes such as the endosomal sorting complex required for transport (ESCRT), as well as associated proteins like ALIX and the tumor susceptibility gene 101 protein (TSG101) [49,50]. ESCRT proteins, including ESCRT-I, ESCRT-II, and ESCRT-III, are crucial for cargo selection and the inward budding process during exosome formation. Some components of the ESCRT machinery, such as vacuolar protein sorting protein 31 (VPS31), vacuolar protein sorting protein 4B (VPS4B), and TSG101, have been identified in endosome-like plasma membrane domains that give rise to exosomes [51]. In addition to the ESCRT-dependent pathway that recognizes ubiquitylated proteins, there are ESCRT-independent mechanisms involved in exosome generation [52]. These unconventional ESCRT-independent pathways appear to be driven by specific lipids, such as ceramides and lysobisphosphatidic acid [53,54].

Exosome Release: The release of exosomes into the extracellular environment involves the transport and docking of MVBs and their subsequent fusion with the plasma membrane [37,55]. Numerous proteins have been implicated in exosome secretion, although the precise mechanism of vesicle release is still not fully understood and likely varies among different cell types. Calcium (Ca^2+^) and pH levels have been proposed to influence exosome release, with evidence suggesting that the process is Ca^2+^-dependent [56] and pH-dependent [57]. Additional mechanisms of exosome secretion are associated with the involvement of the soluble N-ethylmaleimide-sensitive factor attachment protein receptor (SNARE) protein [58]. During this process, the cytoskeleton and cellular contractile machinery collaborate with the SNARE complex to bring opposing membranes together, leading to the detachment of the membrane connection and the subsequent release of the vesicle into the extracellular space [59].

Exosome Uptake: The process of exosome uptake remains a subject of ongoing debate [25], with uncertainties surrounding cell type specificity and the involvement of membrane fusion or endocytosis [24,60]. Additionally, exosome uptake can occur through various mechanisms, including clathrin-mediated endocytosis [61], lipid raft-mediated endocytosis [62], heparan sulfate proteoglycan-dependent endocytosis [63], or phagocytosis [60]. Alternatively, exosomes can be internalized through direct fusion with the plasma membrane [57] or by binding to specific surface molecules on recipient cells, such as phosphatidylserine and lysophosphatidylcholine, and cellular receptors like LFA1, TIM1, and TIM4 (Figure 1) [64].

### 2.2. Exosomes and Cell-to-Cell Communication in the HCC Tumor Microenvironment (TME)

The exchange of exosomes between cancer cells and surrounding non-malignant cells plays a significant role in the progression of cancer [27,65,66]. This signaling network involves various cell types, such as epithelial cells, cancer-associated fibroblasts (CAFs), endothelial cells, neurons, and immune cells, which collectively contribute to either driving or restraining cancer progression [65]. In a state-of-the-art literature review, Tian et al. summarized the role of exosomes in HCC immunotherapy and delineated the role of exosomes in HCC TME [65]. Exosomes derived from tumor cells can directly enter CD8+ T cells, tumor-infiltrating T lymphocytes (TILs), or regulatory T cells (Tregs), leading to inhibition of their anti-tumor function or modulation of their cellular behaviors [67,68,69]. Additionally, HCC-derived exosomes have the ability to activate natural killer (NK) cells, but they can also dampen NK cell cytotoxicity and restrict cytokine secretion [70,71]. Through carrying HCC antigens, HCC-derived exosomes effectively activate dendritic cells (DCs), contributing to anti-tumor immune responses against HCC. Conversely, DC-derived exosomes (DEXs) play a role in reshaping the TME and activating CD8+ T cells [72]. The influence of HCC-derived exosomes on macrophages is dependent on the specific cargo they carry. These exosomes can directly impact the expression of macrophage cell surface receptors and membrane-associated signaling molecules [73,74]. Moreover, they can influence macrophage differentiation, potentially leading to the formation of either M1 (anti-tumorigenic) macrophages [75] or M2 (pro-tumorigenic) macrophages [76]. On the other hand, exosomes originating from macrophages can impact the development of HCC and the activities of other immune cells, as we will delve into further. Notably, high-mobility group box 1 (HMGB1), a DNA-binding nuclear protein, has been identified on exosomal membranes derived from tumors. Recent studies have revealed that HMGB1, carried by exosomes from HCC, plays a role in promoting HCC immune evasion by facilitating the proliferation of TIM-1+ regulatory B cells [77]. Exosomes derived from tumor cells play significant immunoregulatory roles in various cancers, including the regulation of myeloid-derived suppressor cells (MDSCs), a unique cell type in tumor immunity [78]. Unfortunately, there is a lack of relevant studies specifically addressing these aspects of HCC. The concepts mentioned above are visually depicted in Figure 2.

Numerous studies have also examined the relationship between exosomes and non-immune cells in the HCC TME. Due to the hypervascular nature of HCC, exosomes derived from HCC have been found to promote angiogenesis by specifically targeting vascular endothelial cells [79]. Additionally, exosomal CXCR4 has been implicated in promoting lymphangiogenesis in HCC [80]. Exosomes derived from cancer-associated fibroblasts can influence HCC development, invasion, drug resistance, and metabolism [81]. Similarly, adipocyte-derived exosomes exert tumor-promoting effects in HCC [82]. HCC-derived exosomes can be transported to normal hepatocytes or other tumor cells, enhancing their motility [83]. This transport can also stimulate the epithelial-mesenchymal transition of neighboring cells, further promoting HCC invasion [84]. Conversely, exosomes derived from mesenchymal stem cells have shown the ability to partially inhibit the malignant behavior of HCC by transporting specific microRNAs, suggesting their potential utility in HCC treatment [85].

It is important to note that exosome trafficking is notably impacted by factors beyond cell-cell interconnection. These abiotic influences consist of the composition of the extracellular matrix (ECM) that regulates stromal stiffness, the TME’s acidity and hypoxia, as well as various metabolites [86,87]. HCC TME is highly intricate and complex [48,88,89]. Lactate accumulation due to enhanced glycolytic activity under hypoxic conditions acidifies the TME and has direct immunosuppressive effects on immune cells. Targeting lactate-producing enzymes or lactate transporters may offer potential therapeutic approaches [86,90]. Targeting amino acids such as glutamine, arginine, and tryptophan holds promise for modulating tumor progression and immunity within the TME. Inhibiting glutaminolysis may suppress oxidative stress in cancer cells and polarize TAMs into anti-inflammatory M1-like states. Manipulating arginine metabolism can influence myeloid cell functions, while blocking indoleamine 2,3-dioxygenase (IDO1) or reducing kynurenine levels may enhance immune responses against tumors [87]. Finally, exosomes have the ability to generate ATP and transfer mitochondria to recipient cells, affecting their bioenergetics. The presence of ATP and lactate in the TME may aid exosomes in entering cancer cells, potentially influencing metastasis and targeting cancerous cells [86]. Collectively, HCC is seen as a multidimensional spatiotemporal ecosystem where cancer cells act as invasive species and metastasis represents ecological dispersal. Ecological principles, such as communication, competition, predation, parasitism, and mutualism, aid in comprehending HCC development, and exosomes are mediators in these processes [91].

### 2.3. Exploring Exosomes as Efficient Drug Carriers: Advantages and Strategies for Loading Small-Molecule Drugs

Exosomes have emerged as promising drug carriers due to their natural ability to transport genetic material between cells and their targeting capabilities [22]. In comparison to other nanocarriers like liposomes and polymeric nanoparticles, exosomes present numerous advantages, including superior biocompatibility, increased stability, reduced immunogenicity, and the capability of direct drug delivery to cells [92]. Exosomes can specifically target organs due to their unique protein composition and lipid content [93]. Exosomes have shown potential in various therapeutic applications. For example, exosomes loaded with patient-specific neoantigens have been used in cancer immunotherapy, inhibiting tumor growth more effectively than liposomes [94]. Moreover, exosomes carrying drugs such as doxorubicin have shown promise in preventing post-cataract surgery complications by precisely delivering the drug to the target cells [95].

Loading small-molecule drugs into exosomes can be achieved through various strategies. One method is incubation, where exosomes and drugs are co-incubated, allowing the drugs to be loaded into the exosomes. However, the loading efficiency of this method is relatively low. Hence, to enhance the loading of small-molecule drugs, the incubation method is often supplemented with other techniques [22]. Electroporation is an alternative technique that employs an external electric field to temporarily disrupt the exosome membrane, facilitating the diffusion of small-molecule drugs inside [96]. Electroporation has shown high loading efficiency but requires optimization of the electroporation conditions [97]. Sonication uses ultrasound to deform the exosome membrane, enabling drug entry. This method has demonstrated good drug loading efficiency but may affect the structure of exosomes [98]. Extrusion involves passing a mixture of drugs and exosomes through porous membranes, disrupting the exosome membrane, and facilitating drug loading. This method has been successful in loading drugs such as paclitaxel into exosomes [99]. Lastly, the chimeric exosome method combines the advantages of exosomes and liposomes, increasing the loading capacity and half-life of exosomes while minimizing the cytotoxicity associated with liposomes [100].

In conclusion, exosomes offer several advantages as drug carriers, including their natural intercellular transfer ability and targeting capabilities. Loading small-molecule drugs into exosomes can be achieved through various strategies such as incubation, electroporation, sonication, extrusion, and chimeric exosome methods. Each method has its advantages and limitations, and further research is needed to optimize loading efficiency [22]. Nonetheless, exosomes hold great potential as therapeutic agents for targeted drug delivery.

## 3. Macrophages as Targets of Exosomes in HCC

The advancements in immunotherapies for HCC have opened new possibilities for curative treatments [48,65,88,101]. However, challenges persist in the field, and understanding the role of exosomes in the TME of HCC could help overcome these obstacles. Exosomes, originating from various cells in the HCC TME, form a complex network of intercellular communication that can either promote tumor progression or exert anti-tumor effects. These exosomes have been found to play a crucial role in immunotherapy resistance by modulating the expression of PD-1/PD-L1 on immune cells or suppressing the anti-tumor function of neighboring immune cells [102]. Researchers are exploring strategies to enhance HCC immunotherapies by utilizing exosomes as drug carriers or targeting exosomal PD-L1 production [22]. Exosomes also offer potential in tumor vaccine applications, adoptive cell therapy, and MSC-derived exosome therapy for HCC [103].

Macrophages play a crucial role in the TME of HCC, actively contributing to HCC progression by fostering an immunosuppressive environment [12]. Macrophages are recruited in increased numbers to the liver and modulate the expression of the inhibitory molecule PD-L1, which hampers the immune surveillance by cytotoxic T cells [104]. The presence of TAMs in HCC is associated with enhanced tumor growth. Depletion of macrophages or modulation of their function has been shown to reduce HCC growth. Therefore, the participation of macrophages in the immunosuppressive TME of HCC highlights their vital role in driving disease progression [105]. In their study, Tan et al. investigated the functional role of lysyl oxidase-like 4 (LOXL4) in hepatocarcinogenesis and the establishment of an immunosuppressive microenvironment. They observed increased expression of LOXL4 during liver carcinogenesis in mice fed a choline-deficient, l-amino acid-defined diet. HCC cells were observed to secrete LOXL4, which predominantly localizes within hepatic macrophages via internalization through exosomes. Exposure of macrophages to LOXL4 induces an immunosuppressive phenotype characterized by the upregulation of programmed death ligand 1 (PD-L1) expression, which hampers the function of CD8+ T cells. Injection of LOXL4 promotes macrophage infiltration, accelerates tumor growth, and facilitates immune evasion. The immunosuppressive effects of LOXL4 on macrophages depend on interferon-mediated PD-L1 activation. Clinical analysis reveals a positive correlation between LOXL4 expression in CD68+ cells and PD-L1 levels in human HCC tissue, while high LOXL4 expression in CD68+ cells and low CD8A expression in tumor tissue serve as predictive markers for poor survival in HCC patients [102]. Another study conducted by Lu et al. investigated the mechanisms underlying resistance to anti-PD1 therapy in HCC [106]. They focused on the interaction between tumors and macrophages, specifically examining the amplification of the spatially isolated adenosine pathway. They found that an elevated level of circTMEM181, a circular RNA, was associated with a poor response to anti-PD1 therapy and an unfavorable prognosis in HCC patients. Furthermore, high levels of exosomal circTMEM181 created an immunosuppressive microenvironment, leading to anti-PD1 resistance in HCC. The researchers discovered that exosomal circTMEM181 acted as a sponge for miR-488-3p, resulting in increased CD39 expression in macrophages. This, in turn, activated the ATP-adenosine pathway, impairing CD8+ T cell function and driving resistance to anti-PD1 therapy. The study highlights the clinical significance of exosomal circTMEM181 in HCC and suggests that targeting CD39 on macrophages to inhibit the ATP-adenosine pathway could overcome anti-PD1 therapy resistance in HCC [106]. Analogously, Chen et al. investigated the role of Golgi membrane protein 1 (GOLM1) in regulating the immunosuppressive microenvironment in HCC. They found that GOLM1 is positively correlated with high levels of PD-L1 in TAMs and CD8+ T cell exhaustion. Mechanistically, GOLM1 promoted PD-L1 stabilization and its transport into TAMs via exosomes. Combining zoledronic acid with anti-PD-L1 therapy reduced PD-L1+ TAM infiltration and alleviated CD8+ T cell suppression, providing a potential strategy to enhance the efficacy of anti-PD-L1 therapy in HCC [107].

Understandably, macrophages play a crucial role in HCC development by promoting an immunosuppressive environment and modulating the expression of PD-L1, inhibiting cytotoxic T cell activity [102]. Another mechanism involves the presence of exosomal circTMEM181, which leads to an immunosuppressive microenvironment and resistance to anti-PD1 therapy in HCC [106]. Additionally, GOLM1 positively correlates with PD-L1 levels in TAMs and CD8+ T cell exhaustion, with GOLM1 promoting PD-L1 stabilization and transport via exosomes. Combining zoledronic acid with anti-PD-L1 therapy reduces PD-L1+ TAM infiltration and alleviates CD8+ T cell suppression, potentially enhancing therapy efficacy in HCC [107].

Another contribution of exosomes is the regulation of macrophage polarization within the immune HCC TME. 

### 3.1. The Role of Exosomes in M1 Macrophage Polarization

Typically, when M1 macrophages are activated, they display inflammatory properties [12]. In the presence of lipopolysaccharides (LPS) and interferon-gamma (IFN-γ), macrophages tend to polarize in this pro-inflammatory direction. M1 macrophages are known to generate proinflammatory cytokines like interleukin (IL)-12, which can stimulate the proliferation and function of effector T-cells [108]. Moreover, these macrophages possess potent microbicidal and tumoricidal capabilities through the production of reactive oxygen species (ROS) and nitric oxide synthase (iNOS; NOS2). This enzymatic activity aids in the conversion of arginine into nitric oxide (NO) and citrulline, contributing to their antimicrobial and anticancer properties [108]. There is a growing body of evidence suggesting a significant association between exosomes and M1 polarization within the HCC TME.

A recent study investigated the role of miR-142-3p in HCC caused by HBV infection [109]. Hu et al. established an in vitro model by co-culturing HepG2 cells and M1 macrophages, followed by HBV infection. The expression of miR-142-3p was found to be significantly elevated in individuals with HBV-infected HCC and HBV-infected M1-type macrophages. Suppression of exosomal miR-142-3p or upregulation of SLC3A2 resulted in the reversal of ferroptosis and the inhibition of proliferation, migration, and invasion in HCC cells. SLC3A2, also known as solute carrier family 3 member 2 or CD98, is a transmembrane glycoprotein that plays a crucial role in amino acid transport [110]. This study suggests that exosomal miR-142-3p promotes ferroptosis in HBV-infected M1-type macrophages through SLC3A2, influencing the production of GSH, MDA, and Fe2+ and contributing to the development of HCC. Investigating the regulation of miR-142-3p and its target genes can contribute to our understanding of the development of HCC caused by HBV infection and identify potential therapeutic targets for HCC treatment [109]. Hu et al. took a step further in the elucidation of the mechanism by which exosomes derived from HBV-positive HCC cells induce ferroptosis in M1 macrophages [111]. In exosomes obtained from the peripheral blood of HBV-positive HCC patients, there was a significant increase in miR-142-3p expression. M1 macrophages displayed abnormal expression levels of genes associated with intracellular iron metabolism and homeostasis, including ferritin heavy chain 1 (FTH1), transferrin receptor 1 (TfR1), recombinant glutathione peroxidase 4 (GPX4), and activating transcription factor 4 (ATF4). When treated with M1 macrophages, HBV-positive HCC exosomes exhibited a reduced inhibitory effect on HCC cell invasion. However, the impact of HBV-positive HCC exosomes on HCC cells could be reversed by ferroptosis inhibitors. Moreover, the invasive ability of liver cancer cells was weakened upon knockdown of miR-142-3p expression [111]. Collectively, they provided evidence that exosomal miR-142-3p derived from HBV-positive HCC cells promotes liver cancer progression by inducing ferroptosis in M1 macrophages.

Moreover, iron oxide nanoparticles (IONs) have been shown to promote M1 macrophage polarization [112]. Chen et al. conducted a study to investigate the possibility of utilizing exosomes as carriers in combination with PEGylated iron oxide nanoparticles loaded with chlorin e6 (PIONs@E6). The objective was to enhance the immune response against HCC by promoting the polarization of M1 macrophages. They found that PION-contained exosomes stimulated M1 macrophage polarization. The presence of PIONs and iron in the exosomes increased the levels of IL-12 and TNF-α, as well as the expression of CD9 and CD63 on macrophages. Additionally, PIONs induced the generation of ROS in macrophages, further promoting M1-like macrophage polarization. The degree of iron content in the exosomes correlated with enhanced M1 macrophage polarization and ROS levels in a dose-dependent manner, leading to M1-like macrophage polarization and inhibition of tumor growth. In a mouse model of HCC, pretreatment with PION-contained exosomes resulted in reduced tumor volume compared to natural exosomes or PIONs alone. Analysis of macrophage polarization in the tumor microenvironment showed increased levels of IL-12 and TNF-α. Overall, the study highlighted the potential of synergistic effects between exosomes and PIONs@E6 in enhancing immunity against hepatocellular carcinoma by promoting M1 macrophage polarization and reducing tumor growth.

### 3.2. The Role of Exosomes in M2 Macrophage Polarization

The activation of macrophages by IL4, IL-10, and IL-13 in vitro leads to the development of macrophages with immunosuppressive characteristics [12,104]. M2 macrophages secrete various substances, including interleukin-10 (IL-10), transforming growth factor beta (TGF-β), and multiple members of the chemokine (C-C motif) ligand (CCL) family, such as CCL17, CCL18, CCL22, and CCL24. Furthermore, M2 macrophages demonstrate increased expression of PD-L1 [113]. M2 macrophages play a crucial role in initiating the Th2 immune response, which facilitates processes like angiogenesis, tissue remodeling, and repair [113]. Exosomes have been linked to M2 macrophages in HCC. Studies have demonstrated a significant association between exosomes and the polarization of M2 macrophages within the context of HCC [114]. Following IL-6 treatment, exosomal miR-143-3p levels were increased in MHCC-97H and HCCLM3 cell lines. In co-cultured TAMs with high-metastatic-potential HCC cells, MARCKS, a target gene of miR-143-3p, was up-regulated. MARCKS expression was significantly correlated with poorer overall survival (OS) and progress-free survival (PFS). MARCKS demonstrated positive associations with T follicular helper cells (TFH), T helper type 2 cells (Th2), macrophages, T helper type 1 cells (Th1), T cells, NK CD56^bright^ cells, and immature DC (iDC), while being negatively associated with T helper 17 cells (Th17). Additionally, MARCKS potentially influenced M2 polarization and immune evasion [114].

#### 3.2.1. Metabolism Regulation: Harnessing Exosomes for M2 Polarization in Immune Response

Metabolism plays a crucial role in macrophage polarization in the TME [115]. Several metabolic pathways have been identified as important in M2 polarization in HCC, including NAD+ metabolism [116], glutamine metabolism [115], and mitochondrial oxidative phosphorylation (OXPHOS). M2 macrophages obtain energy from mitochondrial OXPHOS [117]. It is important to note that macrophage polarization is a dynamic and tissue-specific process that may not be fully described by a static vision of M1-M2 polarization adopted from in vitro experiments [117].

Ji et al. investigated the role of exosomal ZFPM2-AS1, a long non-coding RNA (lncRNA) gene that is located on chromosome 8, in HCC [118]. Their findings suggested that ZFPM2-AS1 functions as an oncogene in HCC, affecting cell stemness, glycolysis, macrophage polarization, and recruitment. Overexpression of ZFPM2-AS1 was associated with advanced TNM stage, lymphatic metastasis, and poor prognosis in HCC patients. The study highlighted the role of exosomes in intercellular communication and the regulation of HCC progression. They demonstrated that ZFPM2-AS1 interacts with miR-18b-5p, triggering pyruvate kinase M (PKM) expression and modulating glycolysis. Exosomal ZFPM2-AS1 promoted M2 polarization of macrophages and enhanced HCC growth, metastasis, and infiltration. The study suggests that ZFPM2-AS1 could serve as a potential biomarker for HCC detection and treatment. It also emphasizes the importance of exosomal lncRNAs in HCC progression and highlights the potential of exosomes as targets for tumor therapy [118]. Ye et al. aimed to understand the role of the lncRNA miR4458HG in glucose metabolism and the polarization of TAMs. They documented that miR4458HG had significant effects on HCC cell proliferation, glycolysis pathway activation, and the promotion of TAM polarization. Mechanistically, miR4458HG interacted with IGF2BP2, a key protein involved in RNA modifications, leading to the stabilization of target mRNAs associated with glycolysis, such as hexokinase 2 (HK2) and Solute Carrier Family 2 Facilitated Glucose Transporter Member 1 (SLC2A1). Interestingly, the study also found that HCC-derived miR4458HG could be packaged into exosomes, which contributed to the polarization of TAMs by increasing ARG1 expression. These findings highlight the oncogenic nature of miR4458HG in HCC and suggest its potential as a therapeutic target, particularly for HCC patients with altered glucose metabolism [119]. The study emphasizes the importance of understanding the role of exosomes in mediating tumor microenvironment interactions and suggests that further exploration of exosome-mediated communication could provide insights into effective treatment strategies for HCC patients. An illustration of the above-mentioned mechanisms is presented in Figure 3.

#### 3.2.2. Epigenetic Modifiers: Key Players in M2 Polarization in HCC

Epigenetic alterations influence gene regulation by modifying the DNA structure, and their presence has been detected in various cancers, including HCC. Epigenetic modifiers, including microRNAs (miRNAs), long non-coding RNAs (lncRNAs), and circular RNAs (circRNAs), have been recognized as important therapeutic targets for HCC [120]. Mounting evidence suggests that epigenetic modifiers serve as cargo within exosomes, playing a crucial role in the M2 polarization of HCC. Table 1 summarizes the essential information related to the topic.

## 4. Exploring the Potential of Macrophage Exosomes in HCC Cell Targeting

Macrophage exosomes have shown potential for targeting HCC cells. Mounting evidence supports that macrophage-derived exosomes modulate various cellular processes, metabolic pathways, and signaling pathways and influence epigenetic modifications in HCC cells.

### 4.1. Unveiling the Metabolic Landscape: Macrophage-Derived Exosomes Fueling Metabolic Alterations in HCC Cells

Macrophage-derived exosomes have been shown to play a role in promoting HCC cells and regulating their metabolic states. Xu et al. demonstrated that TAMs enhance aerobic glycolysis and proliferation in HCC cells by transmitting a myeloid-derived lncRNA called M2 macrophage polarization-associated lncRNA (lncMMPA) through extracellular exosomes. Functionally, lncMMPA not only polarized M2 macrophages but also acted as a microRNA sponge, interacting with miR-548s and increasing ALDH1A3 mRNA levels, thereby promoting glucose metabolism and cell proliferation in HCC. Additionally, lncMMPA enhanced HCC cell proliferation through its interaction with miR-548s in vivo. Clinically, lncMMPA expression correlates with glycolysis in TAMs and reduced survival in HCC patients. These results underscore the critical significance of lncMMPA in governing HCC and driving metabolic reprogramming via the miR-548s/ALDH1A3 pathway [130].

### 4.2. The Influence of Macrophage-Derived Exosomes in Various HCC Cellular Processes

Recent studies have shown that exosomes derived from macrophages can play a role in various cellular processes in HCC cells. TXNIP (thioredoxin-interacting protein) is implicated in the regulation of cellular processes in HCC cells through its interaction with M2 exosomal miR-27a-3p. Li et al. suggested that miR-27a-3p derived from M2 exosomes targets and downregulates the expression of TXNIP in HCC cells. This implies that M2 exosomal miR-27a-3p acts as a regulatory factor that inhibits the production or function of TXNIP. TXNIP is known to play various roles in cellular functions, including apoptosis, oxidative stress regulation, metabolism, and tumor suppression. Therefore, the suppression of TXNIP expression by M2 exosomal miR-27a-3p may disrupt these normal cellular processes, leading to the promotion of cancer stemness, proliferation, drug resistance, migration, invasion, and tumorigenicity in HCC cells [131]. Chen et al. investigated the role of macrophage-derived exosomes in HCC, focusing on the modulation of exosomal miRNAs by IL-2 [132]. TAMs were isolated from HCC tissues, and their exosomes were treated with IL-2 (ExoIL2-TAM) or left untreated (ExoTAM). They documented that treatment with ExoIL2-TAM led to reduced cell proliferation and metastasis, as well as increased apoptosis, compared to ExoTAM treatment. Furthermore, a specific miRNA, miR-375, was found to be significantly upregulated in ExoIL2-TAM and HCC cells treated with ExoIL2-TAM. This study provides insights into the mechanisms by which IL-2 inhibits HCC and highlights the potential clinical importance of exosomal miRNAs released by TAMs [132]. Regarding HCC stemness, Wang et al. conducted a study to explore the potential of exosomes derived from TAMs in modulating stem cell properties in HCC [133]. They demonstrated that TAM-derived exosomes promote HCC cell proliferation and enhance stem cell properties. Analysis of miRNA profiles revealed significantly lower levels of miR-125a and miR-125b in TAM-derived exosomes and cell lysates. Functional investigations have shown that the administration of exosomes from TAM or the introduction of miR-125a/b into HCC cells resulted in the inhibition of cell proliferation and stem cell characteristics. This effect was achieved by targeting CD90, which serves as a stem cell marker in HCC [133]. These findings highlight the important role of miR-125a/b in targeting CD90 and its impact on cancer stem cells in HCC.

M2 macrophages secrete exosomes containing various cytokines, which play a role in tumor development. Tian et al. aimed to investigate the impact of miR-660-5p-modified M2-derived exosomes (M2-Exo) on HCC development through the regulation of Kruppel-like factor 3 (KLF3) [134]. Higher levels of miR-660-5p and lower levels of KLF3 were observed in HCC tissues. It was found that miR-660-5p targets KLF3. Upregulated miR-660-5p-loaded M2-Exo promoted the growth and epithelial-mesenchymal transition (EMT) of HCC cells, which could be counteracted by overexpressing KLF3. In addition, miR-660-5p-loaded M2-Exo enhanced the tumorigenic ability of HCC cells in mouse models. Conversely, downregulating miR-660-5p attenuated the M2-Exo-mediated promotion of HCC cell growth in both in vitro and in vivo settings. These findings suggest that miR-660-5p-loaded M2-Exo plays a role in augmenting HCC development through the regulation of KLF3. The study provides insights into the contribution of M2 macrophages and their exosomes to HCC progression and highlights the potential therapeutic implications of targeting this pathway [134].

### 4.3. Impact of Macrophage-Derived Exosomes on Signaling Pathways in HCC

Macrophage-derived exosomes (Mφ-Exo) have a diverse impact on tumor initiation, progression, and metastasis, including HCC. Zhang et al. investigated the influence of exosomes derived from macrophages overexpressing recombination signal binding protein for immunoglobulin kappa J region (RBPJ) on HCC [135]. Functional analyses demonstrated that RBPJ+/+ Mφ-Exo and hsa_circ_0004658 inhibited HCC cell proliferation and promoted apoptosis, while knockdown of hsa_circ_0004658 stimulated cell proliferation and migration and inhibited apoptosis in vitro. Furthermore, in vivo studies using a nude mouse xenograft model confirmed the tumor-suppressive effects of RBPJ+/+ Mφ-Exo and hsa_circ_0004658. Mechanistically, hsa_circ_0004658 acted as a competing endogenous RNA (ceRNA) for miR-499b-5p, leading to the upregulation of JAM3. These findings suggest that exosomal hsa_circ_0004658 secreted by RBPJ+/+ Mφ inhibits HCC progression through the hsa_circ_0004658/miR-499b-5p/JAM3 pathway. Additionally, hsa_circ_0004658 may serve as a diagnostic biomarker and potential therapeutic target for HCC [135].

The androgen receptor (AR) plays a significant role in the regulation of HCC initiation and progression [136]. However, the connection between AR and the surrounding macrophages and their impact on HCC progression remains unclear. Liu et al. demonstrated that macrophages may alter the expression of miR-92a-2-5p in exosomes to decrease AR expression in liver HCC cells, consequently promoting their invasive potential [137]. Mechanistic analysis revealed that miR-92a-2-5p in exosomes targets the 3′ untranslated region (3′UTR) of AR mRNA, leading to the suppression of AR translation and subsequent modulation of the PHLPP/p-AKT/β-catenin signaling pathway, resulting in increased invasion of HCC cells. Preclinical investigations demonstrated that inhibition of miR-92a-2-5p effectively suppressed HCC progression by targeting the PHLPP/p-AKT/β-catenin pathway [137]. These findings highlight the role of macrophages and their exosomal miR-92a-2-5p in regulating HCC and suggest that targeting the macrophages/exosomes-miR-92a-2-5p/AR/PHLPP/p-AKT/β-catenin signaling pathway could serve as a promising strategy for the development of novel treatments to effectively suppress HCC progression.

M1 macrophages function as tumor suppressors within the TME. Wang et al. conducted a study to explore the role of circFUT8 in HCC and its interplay with exosomes, M1 macrophages, and m6A modifications. Their findings revealed that circFUT8 was upregulated in HCC cells and facilitated HCC cell growth. Exosomes released by M1 macrophages were observed to deliver miR-628-5p to HCC cells, resulting in the downregulation of human methyltransferase-like 14 (METTL14) expression. METTL14, in turn, facilitated the m6A modification of circFUT8 and enabled its transportation from the nucleus to the cytoplasm. In the cytoplasm, M1 macrophages regulated the circFUT8/miR-552-3p/CHMP4B pathway, effectively inhibiting the progression of HCC. In conclusion, exosomal M1-derived miR-628-5p inhibited the m6A modification of circFUT8, leading to the suppression of HCC development [138].

In summary, macrophage-derived exosomes exert multifaceted roles in HCC, impacting signaling pathways and influencing tumor initiation, progression, and metastasis. Understanding the mechanisms involved in these interactions holds promise for the development of novel diagnostic tools and therapeutic interventions for HCC.

## 5. Discussion

HCC is known to have a unique immune TME [48,88,90,139]. While targeting macrophages in the treatment of HCC holds promise, it is true that macrophage-targeted therapies have faced challenges and limitations in their effectiveness [75]. Various agents targeting TAMs, such as CCR2 inhibitors, anti-CSF1R antibodies, and anti-CD40 agonists, have shown promise in clinical trials for their ability to inhibit macrophage recruitment and eliminate immunosuppression, but there are important drawbacks that need consideration [140]. These drawbacks include off-target effects and side effects associated with their use in cancer therapy [141]. For instance, the cessation of anti-CCL2 treatment has been observed to enhance tumor angiogenesis and metastasis due to rebounding monocyte recruitment [142]. Similarly, discontinuation of CSF-1R blockade can lead to tumor recurrence through the accumulation of monocyte-derived macrophages [143]. Additionally, systemic depletion of TAMs using clodronate-encapsulated liposomes has demonstrated tumor growth suppression, but indiscriminate clearance of antitumor macrophages may exacerbate tumor progression [144]. Given these limitations, there is an urgent need to develop more precise and specific strategies targeting TAMs for HCC treatment. Moreover, considering the potential clinical application of microRNA delivery, utilization of macrophage-derived exosomes has been suggested as a promising candidate for tumor therapy by selective targeting of TAMs.

Indeed, exosomes have emerged as promising vehicles for nanotheranostics in HCC due to their unique properties, such as their natural biocompatibility, stability, and ability to carry various cargo molecules [145]. Nanotheranostics refers to the integration of diagnostic and therapeutic functionalities into a single nanoscale system. In the context of HCC, nanotheranostics offer potential advantages for precise imaging, targeted therapy, and personalized medicine. Various types of nanoparticles, such as quantum dots, gold nanoparticles, and iron oxide nanoparticles, can be engineered to carry both imaging agents and therapeutic payloads [146,147,148]. These nanosystems enable non-invasive imaging of HCC lesions, including early-stage tumors and metastases, providing real-time information for accurate diagnosis and monitoring of treatment response [149,150]. Furthermore, nanotheranostics can be designed to selectively accumulate in HCC cells or the tumor microenvironment, allowing for targeted delivery of therapeutic agents, such as chemotherapy drugs or gene therapies, directly to the tumor site while minimizing off-target effects [150]. Additionally, nanotheranostics can be tailored to respond to specific tumor characteristics or stimuli, such as pH, temperature, or enzymatic activity, enabling controlled release of therapeutics within the ΤΜΕ [151,152]. Overall, nanotheranostics holds great promise in advancing the field of HCC treatment by combining diagnostics and therapeutics into a single nanoscale platform for enhanced precision, efficacy, and personalized care.

Despite the lack of clinical studies regarding the role of exosomes in HCC, mounting evidence about the role of exosomes in several gastrointestinal (GI) cancers has started to emerge [153,154]. Lapitz et al. conducted a thorough investigation into liquid biopsy-based protein biomarkers for CCA. CCA is known for its challenging prognosis and lack of early diagnostic methods, particularly in individuals with primary sclerosing cholangitis (PSC). They focused on serum EVs as potential carriers of informative protein biomarkers. They carefully characterized EVs from various patient groups, including isolated PSC, concomitant PSC-CCA, PSC patients who developed CCA during follow-up, CCAs of non-PSC origin, as well as HCC cases and healthy individuals. Mass spectrometry and ELISA analyses identified diagnostic biomarkers specifically for PSC-CCA, non-PSC CCA, and Pan-CCAs with potential clinical value for improving the differential diagnosis between intrahepatic CCA and HCC. They have also found key biomarkers, such as CRP/FIBRINOGEN/FRIL, that could distinguish PSC-CCA from isolated PSC with high accuracy. Combining these biomarkers with carbohydrate antigen 19-9 further enhanced diagnostic precision. Moreover, the combination of CRP/PIGR/VWF facilitated the distinction between LD non-PSC CCAs and healthy individuals, while CRP/FRIL provided precise diagnosis for LD Pan-CCA cases. The study also provided insights into the predictive capacity of certain biomarkers, including CRP/FIBRINOGEN/FRIL/PIGR, for CCA development in PSC patients before clinical evidence of malignancy emerged. These findings offer promising potential for early detection and risk assessment in CCA. Transcriptomic analysis indicated that the identified serum EV biomarkers were predominantly expressed in hepatobiliary tissues. Single-cell RNA sequencing and immunofluorescence analysis of CCA tumors confirmed their presence mainly in malignant cholangiocytes, supporting the value of serum EV biomarkers as liquid biopsy tools for CCA diagnostics and prognostication. In conclusion, this study offered valuable evidence of liquid biopsy-based protein biomarkers for the early diagnosis, risk prediction, and prognostication of CCA [153]. As the field of liquid biopsy research continues to develop, there is potential for transformative advancements in the diagnosis and treatment of various cancers, including both CCA and HCC. Guo et al. identified EV-derived GClnc1 as a promising circulating biomarker for the early detection of gastric cancer (GC) [154]. GClnc1 was found to be up-regulated in both tissue and circulating EV samples in early-stage GC (stages I and II), with a high area under the curve (AUC) of 0.9369 (95% CI: 0.9073–0.9664). EV-derived GClnc1 effectively distinguished early-stage GC from precancerous lesions (chronic atrophic gastritis and intestinal metaplasia) and GC with negative traditional gastrointestinal biomarkers (CEA, CA72-4, and CA19-9). GClnc1 demonstrated low levels in postsurgery and other gastrointestinal tumor plasma samples, highlighting its specific association with GC. Collectively, the identification of specific diagnostic and prognostic biomarkers in serum EVs represents a significant step towards personalized medicine and improved outcomes for GI cancer patients.

A number of limitations and challenges for the therapeutic utilization of exosomes in HCC face several limitations and challenges. The heterogeneity of exosomes, both in terms of their cargo composition and cellular origin, remains a major limitation. Variability in cargo composition among exosomes derived from different cell types can affect their therapeutic efficacy and specificity [22]. Another limitation is the complex interplay between exosomes and the tumor microenvironment in HCC. Exosomes participate in cell-to-cell communication and can modulate various processes involved in tumor progression, including angiogenesis, immune evasion, and drug resistance. However, their interaction with recipient cells in the tumor microenvironment can be multifaceted and context-dependent. Understanding the precise mechanisms of exosome-mediated communication in HCC is crucial for optimizing their therapeutic application and minimizing potential unintended effects [155]. Technical challenges such as scalability and standardization of exosome production pose additional limitations. Large-scale production of exosomes with consistent quality and quantity remains a hurdle. Purification methods for isolating exosomes need to be refined to ensure the removal of unwanted contaminants and maximize therapeutic efficacy. Standardization of exosome-based therapies is essential for regulatory approval and widespread clinical implementation [22]. Additionally, the therapeutic utilization of exosomes in HCC is mainly supported by preclinical studies. Although promising results have been observed in preclinical models, it is important to note that the translation of exosome-based therapies to clinical applications is still in the early stages, and only one clinical trial (NCT05575622) is currently being conducted. The objective of this study is to assess the effectiveness of immunotherapy in HCC patients by analyzing both circulating tumor cells (CTC) and exosomes. Liquid biopsy methods will be employed to monitor PD-L1 expression on CTC and exosomes, as well as detect exosomal LAG-3 protein in the peripheral blood of HCC patients. The study will include 200 participants with HCC who are undergoing immunotherapy, with peripheral blood samples collected at various stages of treatment. Tumor response evaluation will be based on RECIST criteria, and clinical data will be gathered accordingly. Primary outcome measures will focus on CTC-PD-L1, exosomal PD-L1, and exosomal LAG-3 levels throughout the treatment period and follow-up. In summary, the therapeutic utilization of exosomes in HCC is limited by the heterogeneity of exosome cargo, complex interactions within the tumor microenvironment, technical challenges in production and purification, and safety considerations. Addressing these limitations will be crucial to harnessing the full potential of exosome-based therapies for HCC treatment. 

## 6. Conclusions

The utilization of exosomes in HCC holds promising future perspectives. Exosomes have emerged as valuable tools in cancer research and therapy due to their unique properties, including their ability to transport various biomolecules and their potential as diagnostic and therapeutic vehicles. In the context of HCC, exosomes offer several potential applications. The utilization of exosomes in HCC presents exciting future perspectives for early detection, personalized therapy, and modulation of the immune response. Continued research and development efforts are needed to overcome the current limitations and harness the full potential of exosomes for improved HCC management.

## Figures and Tables

**Figure 1 cells-12-02036-f001:**
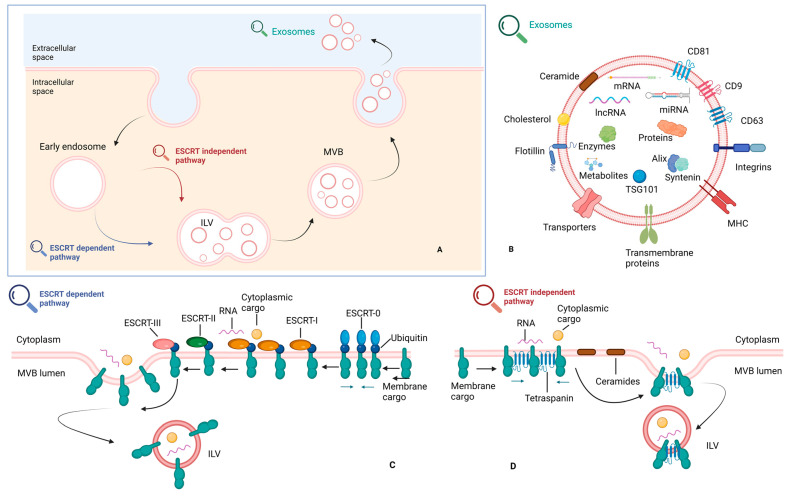
The fundamental steps of exosome biogenesis, the main classes of exosome cargos, and the basic molecules of the formation machinery: (**B**) EVs comprise exosomes and microvesicles, with exosomes ranging from 40 to 150 nm and microvesicles varying from 50 to 1000 nm. EVs contain tetraspanins, integrins, Alix, TSG101, and flotillin, with exosomes enriched in CD63, CD9, CD81, Alix, and syntenin. (**A**) Exosome biogenesis involves inward budding within endosomes, leading to MVBs and subsequent release into the extracellular space. Exosome cargo includes proteins, lipids, and functional RNA molecules. (**C**,**D**) Cargo sorting involves ESCRT-dependent and ESCRT-independent pathways. Exosomes are released via fusion of MVBs with the plasma membrane, and uptake can occur through endocytosis, fusion, or receptor-mediated binding. Created with Biorender.com (accessed on 2 August 2023).

**Figure 2 cells-12-02036-f002:**
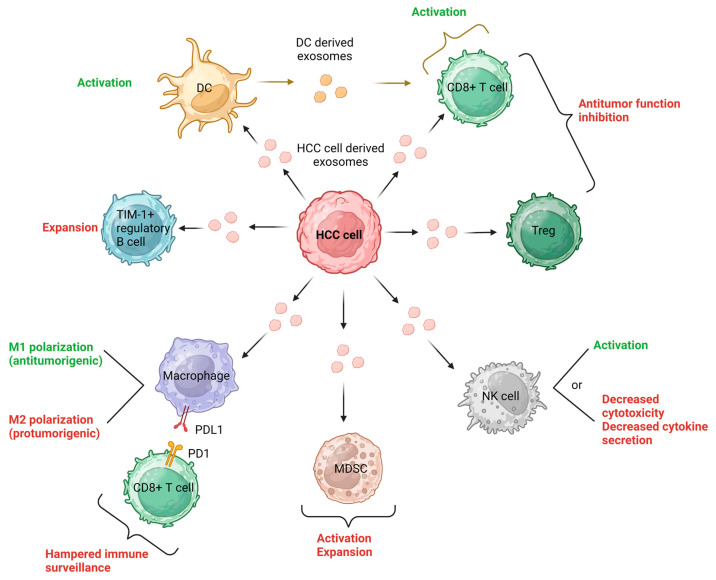
Exosomes play a significant role in cell-to-cell communication within the TME of HCC. HCC-derived exosomes affect immune cells, inhibiting their anti-tumor function or modulating their behaviors. They also activate dendritic cells and impact macrophage differentiation, while exosomal HMGB1 promotes HCC immune evasion. These findings indicate the complex role of exosomes in HCC pathogenesis. Green: antitumorigenic responses, Red: protumorigenic responses. Created with Biorender.com (accessed on 2 August 2023).

**Figure 3 cells-12-02036-f003:**
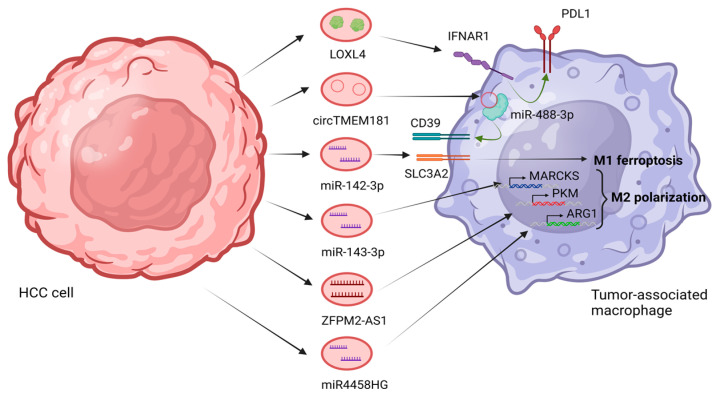
Exosomes derived from HCC cells can target macrophages and influence their function. For instance, exosomal LOXL4 promotes an immunosuppressive phenotype in macrophages, leading to immune evasion and tumor growth. Exosomal circTMEM181 derived from HBV-positive HCC cells induces an immunosuppressive microenvironment and resistance to anti-PD1 therapy by activating the ATP-adenosine pathway in macrophages. Additionally, exosomal GOLM1 enhances PD-L1 expression in macrophages, contributing to CD8+ T cell suppression. Exosomes can also regulate macrophage polarization within the TME. Exosomes from high-metastatic-potential HCC cells influence M2 macrophage polarization by modulating target gene expression. Furthermore, exosomal lncRNAs affect macrophage polarization and glycolysis, promoting HCC growth and metastasis. Understanding the role of exosomes in macrophage targeting and polarization provides insights into HCC progression and potential therapeutic strategies. Created with Biorender.com.

**Table 1 cells-12-02036-t001:** Published studies presenting an in-depth analysis of the role of exosomes in M2 polarization.

Author, Year	Molecule/Type of Study	Mechanisms	Outcomes	Ref.
Wang, 2021	hsa_circ_0074854/preclinical	Interaction with human antigen R (HuR).	Downregulation of hsa_circ_0074854 hinders the migration and invasion of HCC cells.	[121]
		Exosomes containing decreased levels of hsa_circ_0074854 exhibited the ability to inhibit M2 polarization.		
Wang, 2021	LncRNA HMMR-AS1/preclinical	HMMR-AS1 is competitively bound to miR-147a, preventing ARID3A degradation.	Inhibiting HMMR-AS1 expression substantially suppresses tumor growth in vitro and in vivo.	[122]
		Exosomes carrying HMMR-AS1 facilitated M2 polarization.		
		HIF-1α enhanced HMMR-AS1 transcription, leading to an increased secretion of exosomes.		
Li, 2018	lncRNA TUC339/in vitro	Tumor-derived exosomes containing elevated levels of the lncRNA TUC339 are taken up by THP-1 cells (macrophages).	TUC339 is involved in the regulation of macrophage activation and polarization, specifically M1/M2 polarization.	[123]
		Suppression of TUC339 in macrophages leads to increased pro-inflammatory cytokine production, enhanced co-stimulatory molecule expression and improved phagocytosis.	TUC339 affects various cellular processes and pathways related to cytokine signaling, chemokine receptor binding, toll-like receptor signaling, phagocytosis, cytoskeleton regulation, and cell proliferation in macrophages.	
		Overexpression of TUC339 in macrophages has the opposite effect, reducing pro-inflammatory cytokine production, co-stimulatory molecule expression, and phagocytosis.		
		TUC339 is involved in cytokine-cytokine receptor interaction, CXCR chemokine receptor binding, Toll-like receptor signaling, FcγR-mediated phagocytosis, regulation of the actin cytoskeleton, and cell proliferation in macrophages.		
Tao, 2022	LncRNA MAPKAPK5_AS1 (MAAS)/preclinical	Upregulation of MAAS in HBV-related HCC cancerous tissues. Promotion of c-Myc-induced transcriptional activation.	Poor survival probability in patients with high MAAS expression. Facilitated proliferation of HBV+HCC cells in vitro and in vivo.	[124]
		Stabilization of the c-Myc protein. Facilitation of the G1/S transition. Enhancing the N6-methyladenosine modification of MAAS mediated by methyltransferase-like 3.	Activation of cyclin-dependent kinase 4 (CDK4), CDK6, and S-phase kinase-associated protein 2. Promotion of cell proliferation in HBV and HCC cells.	
		Transfer of MAAS to HBV + HCC cells via exosomes derived from M2 macrophages.	Transfer of MAAS from M2 macrophages to HBV + HCC cells via exosomes. Establishment of a positive feedback loop between HBeAg, MAAS expression, and M2 macrophages.	
Lv, 2022	lncRNA FAL1/in vitro	Extracellular vesicular lncRNA FAL1 induces macrophage M2 polarization.	Macrophage M2 polarization is promoted by FAL1-enriched EVs. Co-culture of FAL1-overexpressing macrophages with HepG2 cells facilitates the malignant progression of HepG2 cells.	[125]
		FAL1-enriched EVs stimulate the activation of the Wnt/β-catenin signaling pathway in HCC cells.	Activation of the Wnt/β-catenin signaling pathway in HCC cells is observed when co-cultured with EVs-incubated macrophages. Mouse xenograft tumor growth is increased by FAL1-enriched EVs-incubated macrophages.	
Zongqiang, 2022	miR-452-5p/preclinical	HCC cells secrete exosomal miR-452-5p.	Exosomal miR-452-5p promotes the progression of HCC. HCC cell-derived exosomes, along with miR-452-5p overexpression, accelerate HCC migration and invasion.	[126]
		Exosomal miR-452-5p induces polarization of M2 macrophages. MiR-452-5p targets TIMP3, leading to its downregulation.	In vivo experiments demonstrate that miR-452-5p accelerates HCC growth and metastasis. Overexpression of TIMP3 inhibits the pro-invasive and migratory effects of HCC cell-derived exosomes.	
Yu, 2023	miR-21-5p/clinical and preclinical	HCC-derived exosomes mediate intercellular communication and promote TAMs’ phenotypic differentiation into M2-like macrophages.	HCC cell-derived exosomes significantly induce the differentiation of THP-1 macrophages into M2-like macrophages, characterized by increased production of TGF-β and IL-10.	[127]
		Exosomal miR-21-5p is closely related to TAM differentiation and directly targets the 3′-UTR of RhoB in THP-1 cells.	Overexpression of miR-21-5p in THP-1 cells leads to downregulation of IL-1β levels, enhanced production of IL-10, and promotes malignant growth of HCC cells in vitro.	
		Downregulation of RhoB weakens the MAPK signaling pathways in THP-1 cells.	Tumor-derived miR-21-5p facilitates the malignant progression of HCC by mediating intercellular crosstalk between tumor cells and macrophages.	
Liu, 2019	miR-23a-3p/in vitro and preclinical		Exosomes derived from ER-stressed HCC cells stimulate macrophages to upregulate the expression of PD-L1.	[74]
		The induction of ER stress leads to the upregulation of ER stress markers, including glucose-regulated protein 78 (GRP78), activating transcription factor 6 (ATF6), PKR-like endoplasmic reticulum kinase (PERK), and inositol-requiring enzyme 1α (IRE1α), in HCC cells.	Increased PD-L1 expression on macrophages inhibits T-cell function by interacting with the PD-1 receptor on T cells, leading to a decreased CD8+ T-cell ratio, reduced IL2 production, and increased T-cell apoptosis.	
		ER stress induces HCC cells to release exosomes containing high levels of miR-23a-3p.	The release of exosomal miR-23a-3p and subsequent upregulation of PD-L1 on macrophages contribute to the evasion of antitumor immunity by HCC cells.	
Xu, 2022	miR-200b-3p/in vitro and in vivo	HCC cell-derived miR-200b-3p exosomes downregulate ZEB1.	Induction of M2 polarization in macrophages.	[128]
		MiR-200b-3p exosomes upregulate IL4.	Enhanced proliferation and metastasis of HCC cells.	
		Activation of the JAK/STAT signaling pathway in M2 macrophages.	Establishment of a feedback loop between HCC cells and M2 macrophages.	
		Increased expression of PIM1 and VEGFα.	Promotion of tumor growth and progression in the tumor microenvironment.	
Zhao, 2020	miR-934/in vivo and in vitro	Tumor cells release exosomal miR-934.	Induction of M2 macrophage polarization.	[129]
		Exosomal miR-934 is internalized by macrophages.	Activation of the CXCL13/CXCR5/NFκB/p65/miR-934 positive feedback loop.	
		Exosomal miR-934 downregulates PTEN expression.	Promotion of premetastatic niche formation.	
		Downregulation of PTEN activates the PI3K/AKT signaling pathway.	Facilitation of colorectal cancer liver metastasis (CRLM).	
		Polarized M2 macrophages secrete CXCL13.	Correlation of miR-934 overexpression with poor prognosis in CRC patients.	

## Data Availability

Not applicable.

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
