# Peer review of "Unveiling the Yin-Yang Balance of M1 and M2 Macrophages in Hepatocellular Carcinoma: Role of Exosomes in Tumor Microenvironment and Immune Modulation"

_cells, 2023, doi:10.3390/cells12162036_

Round 1

Reviewer 1 Report

Papadakos SP, et al. reviewed the role of exosomes in TME and immunological modulation.

1.      In abstract section, do not use abbreviation: TME without any explanation. Add M1 and M2.

2.      In Introduction section, do not use abbreviation: CCA without any explanation.

3.      In Figure 1, add ALIX.

4.      See the review article about exsosomes and HCC: Sasaki R et al. Int J Mol Sci. 2019 Mar 20;20(6):1406. doi: 10.3390/ijms20061406. PMID: 30897788

Papadakos SP, et al. reviewed the role of exosomes in TME and immunological modulation.

1.      In abstract section, do not use abbreviation: TME without any explanation. Add M1 and M2.

2.      In Introduction section, do not use abbreviation: CCA without any explanation.

3.      In Figure 1, add ALIX.

4.      See the review article about exsosomes and HCC: Sasaki R et al. Int J Mol Sci. 2019 Mar 20;20(6):1406. doi: 10.3390/ijms20061406. PMID: 30897788

Reviewer 2 Report

This review comprehensively provides a better understanding of the role of macrophage-derived exosomes in HCC progression and offer new avenues for targeted interventions and improved patient outcomes. Overall, it is a well-organized paper. Several points should be noted as below.

1) Besides the relationship between exosomes and immune/non-immune cells in the HCC TME, how about the role of abiotic factors on regulation of exosomes, e.g. pH, ECM, stromal stiffness.

2) Figure 2, as to “They are exchanged between cancer cells and non-malignant cells”, however, all of the arrows were all unidirectional. Additionally, in this image, “non-malignant cells” were present with non-immune cells symbols without any other “non-malignant cells” such as endothelial cells, cancer-associated fibroblasts, adipocytes and mesenchymal stem cells.  

3) The relation of cancer and TME was so complex, we can also feel this from the positive and negative feedback regulation between HCC and immune/non immune cells through extracellular vesicles in this review. So, what is the essence behind these complex interactions? A recent paper has gave the possible answer, it proposes that cancer is not a genetic disease but an ecological disease: a multidimensional spatiotemporal "unity of ecology and evolution" pathological ecosystem (https://www.thno.org/v13p1607.htm). This viewpoint may help to add some differences.

Reviewer 3 Report

In the Manuscript by Papadakos SP et al., the authors made a literature review of the role of extracellular vesicles (EVs) in the settlement of Tumor Microenvironment (TME) in Hepatocellular Carcinoma (HCC), focusing on the reciprocal role of HCC cells and Macrophages and EV thereof. After a brief introduction on the general features of EVs, the authors deal with the intercellular EVs-based communication in HHC microenvironment, the possible role of EVs as drug carriers, the macrophages as target of EVs released by HCC and the ability of these EVs to alter M1/M2 polarization with the ultimate goal of establish a pro tumoral TME. In the final part of the MS, the authors report some data on the transfer of epigenetic modifiers (i.e. miRNAs. lncRNAs etc) made by EVs in the context of HCC and on the possible target of these EVs for tumor therapy, especially by evaluating the role of macrophage derived EVs on HCC tumor growth.

The review is well conceived well organized, and it covers most of the aspects connected to the topic that is really relevant not only for HCC but, in general, for the settlement and the development of tumors and metastasis.

I did not find any important critical issues, and I just find the acronym CCA in line 37 that is not explained and must be make explicit. Overall, I consider the MS by Papadakos SP et al. suitable for publication in Cells.

Round 2

Reviewer 2 Report

The authors have fully answered these suggestions.